# Implementation of MEC-Assisted Collective Perception in an Integrated Artery/Simu5G Simulation Framework

**DOI:** 10.3390/s23187968

**Published:** 2023-09-19

**Authors:** Gergely Attila Kovács, László Bokor

**Affiliations:** 1Department of Networked Systems and Services, Faculty of Electrical Engineering and Informatics, Budapest University of Technology and Economics, Műegyetem rkp. 3., H-1111 Budapest, Hungary; bokorl@hit.bme.hu; 2ELKH-BME Cloud Applications Research Group, Magyar Tudósok Körútja 2, H-1117 Budapest, Hungary

**Keywords:** collective perception, multi-access edge computing, 5G/6G, V2X/C-V2X, Simu5G/Artery/OMNeT++ simulations

## Abstract

Advanced vehicle-to-everything (V2X) safety applications must operate with ultra-low latency and be highly reliable. Therefore, they require sophisticated supporting technologies. This is especially true for cooperative applications, such as Collective Perception (CP), where a large amount of data constantly flows among vehicles and between vehicles and a network intelligence server. Both low and high-level support is needed for such an operation, meaning that various access technologies and other architectural elements also need to incorporate features enabling the effective use of V2X applications with strict requirements. The new 5G core architecture promises even more supporting technologies, like Multi-access Edge Computing (MEC). To test the performance of these technologies, an integrated framework for V2X simulations with 5G network elements is proposed in the form of combining Simu5G, a standalone 5G implementation, with the go-to V2X-simulator, Artery. As a first step toward a fully functional MEC-assisted CP Service, an extension to Simu5G’s edge implementation is introduced. The edge application is responsible for dispatching the Collective Perception Messages generated by the vehicles via the 5G connectivity so that a MEC server provided by the network can process incoming data. Simulation results prove the operability of the proposed integrated system and edge computing’s potential in assisting V2X scenarios.

## 1. Introduction

The study of vehicular communications (V2X—Vehicle-to-Everything) is ever-growing as more and more parties, ranging from standardization organizations to end-device manufacturers and the most significant automotive industry representatives, invest great effort in research and development. New and even exotic solutions, like coverage extension through air base stations [1], emerge one after the other to enhance the capabilities of vehicular networking and broaden the scale of available applications and services. V2X, in the end, helps to realize the technologies required for a safer and more effective way of transportation. Undoubtedly, many recent advanced intelligent transport system (ITS) technologies and services are based on cooperative behavior that is foreseen as a prelude to fully automated systems operating around us. The advancements in radio access technologies (RATs) used in V2X scenarios and the constant refinement of the conceptual services that rely on ultra-low latency and very high reliability bring us even closer to the widespread application of Day 2 and even Day 3+ V2X services.

These complex services are likely to require other advanced supporting technologies and trends provided by mobile communication systems. One such enabler for many resource-heavy applications is a paradigm mainly taken up during the development of 5G enhancements called Multi-access Edge Computing (MEC). On the other hand, the connection between V2X and mobile technologies is further increased by well-built RATs. 4G LTE and 5G NR both provide many useful features in the physical and medium access layers that enable V2X.

There are two ways to test the applicability of MEC and other 5G developments in V2X scenarios: field tests with real hardware (cars, network hardware, servers, etc.) or through simulations. Of course, the first option is often expensive and can be time-consuming since there are few widely available implementations of the latest 5G services, if any. Therefore, we aim to expand the already existing V2X simulation ecosystem with the latest 5G updates to create an integrated simulator connecting the two fields. Using open-source libraries and frameworks ensures that the integrated system remains available to the general public and other researchers. Additionally, such a framework makes testing relevant technologies and services more flexible and much quicker by running simulations in different scenarios with a wide range of adjustable parameters. The possibilities include testing whole architecture models against one another or discovering protocol alternatives. A key difference between the proposed tool and many others appearing in recent research studies is that most solutions often do not contain such a wide range of model components and adjustable simulation parameters, whereas our proposal aims to deliver a complete model including network (both ITS-G5/DSRC and 5G-based architectures), vehicle traffic, sensor and environment, and MEC. Such a tool helps research activities on V2X and 5G in general. However, we narrow the scope of our practical research to Collective Perception (CP), a Day 2 (and beyond) V2X service, and the potential of MEC in assisting CP-based applications for cooperative environment sensing, sensor fusion, or Misbehavior Detection.

As an extension of our work originally presented at the 45th International Conference on Telecommunications and Signal Processing [2], this paper provides an insight into the similarities and differences between the two leading access technologies, namely 5G NR-V2X and 802.11bd, a detailed report on the latest updates regarding the Collective Perception Message (CPM) and the architectural support of ETSI MEC for V2X applications, and a potential use case of a 5G-based V2X service leveraging CPM and edge computing. Regarding the practical evaluation work, we progressed with the design of the MEC app using the collected CPMs and the whole system in general according to the Simu5G developments in terms of their MEC implementation. The scenario in the future will also be extended with the deployment of a Roadside Unit (RSU) collecting the CPMs according to the ITS-G5 standards and sending them to a conventional cloud server architecture. This way, we will be able to compare the performance of the Wi-Fi approach to the case utilizing MEC provided by a 5G infrastructure. Our motivation is to assess the current state of the standardization of said technologies and to provide valuable simulation tools for other academic researchers wishing to familiarize themselves with and conduct research using the state-of-the-art.

The remainder of this paper is organized as follows. Section 2 describes the most relevant topics and technologies in connection with our research. In Section 2.1, we introduce and compare the two relevant access technologies used for V2X, along with the enhancements of each technology family. Section 2.2 describes the soon-to-be standardized Collective Perception service, with the basic idea behind the service and the proposed message format. A brief description of ideas for 3GPP infrastructural extensions for V2X services support can be found in Section 2.3. Section 2.4 and Section 2.5 collect general and MEC-assisted use cases for Collective Perception, respectively. Section 3 introduces related works to this study that inspired the simulation scenario’s design. Section 4 details the integrated framework and the simulation scenario where the initial proof-of-concept tests were conducted. In Section 5, we conclude the paper and draw future research directions.

## 2. Background

### 2.1. Relevant V2X Access Technologies

The original idea behind Vehicle-to-Vehicle communication (V2V) was that communicating parties would form Vehicular Ad hoc Networks (VANETs), for which a dedicated radio technology was developed. The access technologies following this idea are Dedicated Short Range Communications (DSRC) [3] and ITS-G5 [4] in the US and Europe, respectively. Both of these technologies rely on the 802.11p standard for the physical (PHY) and medium access control (MAC) layers [5]. 3GPP standardized Cellular-V2X (C-V2X), which enables direct communication between vehicles using its PC5 side-link interface, like DSRC or ITS-G5. Additionally, it can take advantage of the cellular infrastructure using the traditional LTE-Uu interface to gain more resources for specific applications.

Studies have shown that both access technologies fail to provide the required quality of service when widespread V2X deployment with multiple simultaneous cooperative services is considered, especially in terms of latency, with increasing the density of communicating vehicles [5,6]. While DSRC/ITS-G5 is regarded as the most mature technology that seems robust up to multiple hundred meters of range in real deployments, numerical results of LTE-based C-V2X show stability at even greater distances [7]. To overcome the identified weaknesses of the two alternatives and provide higher reliability and lower latency in general, an IEEE task group BD (formerly known as NGV—Next Generation V2X) is working on 802.11bd, an improvement to the legacy 802.11p. IEEE approved the draft standard for 802.11bd at the end of 2022 [8]. Similarly, 3GPP has released its enhancements based on 5G NR called NR-V2X in Rel.16 and plans to improve more in Rel.17 and beyond [9]. Table 1 shows a brief comparison of all four technologies [7,10,11].

The debate for which of the two technologies should be the flagship for V2X is old. Of course, the investors and developers promote their interests, sometimes creating rather fierce competition. As the standards congeal, newer simulation studies can be conducted to have an impression on real-world useability. A few recent studies conclude that despite the impressive improvements in both technologies, NR-V2X might hold immense potential [10,12,13]. Although the advancements of the 802.11-based technology look very promising too, it seems that in the US, where a new policy was issued, some might consider a significant setback for the dedicated short-range radio connectivity for vehicles.

The problem is that despite the advantages regarding traffic safety, DSRC was not gaining enough popularity since the FCC arrogated a 75 MHz wide band in the 5.9 GHz range for the technology. Originally, DSRC was to address communication between vehicles only, but over the years, engineers have planned to widen the scope to contain infrastructural elements or even pedestrians as well. Sadly, according to FCC decision-makers, in the last two decades, no significant systems based on DSRC were implemented to justify the need for such a wide spectrum. There are several pilot deployments (e.g., within the Connected Vehicle TestBeds initiative), and there was an attempt in 2016 by the National Highway Traffic Safety Administration to make DSRC-based V2X mandatory in all new cars and trucks. Still, neither had an unquestionable impact on commercial availability. Audi made the only implementation close to this attempt, but in addition to some apparent flaws, it completely neglected DSRC and was based on 4G LTE.

The FCC decided in November 2020 to reallocate the lower 45 MHz part of the bandwidth previously dedicated to DSRC and give it to widen the unlicensed band used by Wi-Fi. The organization also proposed that the remaining 30 MHz should be utilized by C-V2X rather than DSRC since the market-driven cellular approaches evolve faster. Of course, automotive parties objected and argued that the remaining 30 MHz would not be nearly enough for future developments of DSRC and that the interference from the unlicensed band could hinder the reliability of the safety applications bound to the narrower band. In August 2022, the US Court of Appeals for the District of Columbia Circuit approved the decision, saying that it was justified with enough research and that the FCC has the right to ensure the practical usage of the available radio spectrum [14].

This, in summary, should not mean the sudden end of 802.11-based V2X, but it shows a clear message that, at least in the US, a shift to the cellular approach is to be expected. This might not be a surprise, considering the simulation studies showing that deploying NR-V2X could be more beneficial performance-wise [15,16]. On one hand, it is clear that, as of this day, neither of the two technologies are flawless in all intended scenarios, which should mean that there is still a need for DSRC (or, more precisely, the successor based on 802.11bd). This way of thinking is also supported by current European deployment activities led by the C-ROADS Platform, where Wi-Fi-based ad hoc vehicular communication (ETSI ITS-G5) is still the major access technology to be considered. On the other hand, there is no doubt that the infrastructural advantages that come with 5G NR and the much more rapid development and distribution of network devices prove that it is wise to rethink where V2X is going in the upcoming years. Though this decision is specific to the US, the impact of this on the European ETSI ITS-G5 deployments will be interesting, especially if we consider that China is also massively moving towards C V2X, even more so than the US. This article deals with a 5G-focused scenario where the unique features of Multi-Access Edge Computing can be used from the complete cellular ecosystem of 5G and beyond for advanced V2X service provisioning.

### 2.2. Collective Perception

#### 2.2.1. General Idea

The Collective Perception Service (CPS) is an advanced Day 2 solution standardized by ETSI [17,18] that serves as a basis for advanced collaborative V2X applications for vehicles capable of sensing their environment. These vehicles using CPS will share their sensory information wirelessly, enabling the distributed or centralized processing of the collected data, which can be used in a diverse set of safety-related use cases. This service is a crucial milestone in reaching fully automated cooperative transportation systems. Yet, it should have a massive impact on the number and gravity of traffic accidents involving human drivers. To extend the perception of the environment beyond the range of the respective sensors of each vehicle, the service exchanges data via periodically sent messages. Since a wide variety of sensor types can be present on vehicles, these messages, CPMs, to be exact, are structured explicitly for sharing sensory data effectively [18]. The provisioned information includes the sender’s static and dynamic properties (exact details depend on the sender type, i.e., whether it is an RSU or a vehicle), the type and parameters of each sensor in use, and a list of detected objects with additional confidence values.

The technical report preceding the specification [19] presented, along with the basic idea of the service and the message format, described two general use cases where the CP service can be utilized on its own, detecting non-connected road users and detecting safety-critical objects. In both use cases, the goal is to share information among vehicles about objects or other traffic participants that may not be visible to some, e.g., because of non-line-of-sight. Figure 1 illustrates how the number of perceived vehicles can expand using CPS [20] and how it affects the functioning of the local perception database of a vehicle.

With automated vehicles, the knowledge gathered by collectively perceiving the environment will be the basis for complex algorithms that, e.g., calculate normal and emergency trajectories. The report also considered the integration of information received through Cooperative Awareness Messages (CAM), the messages used by the Day 1 Cooperative Awareness (CA) service, further enhancing the quality of different services provided by the infrastructure. One example could be a central entity aggregating the information gathered from multiple CAM and CPM sources near an intersection and resending relevant information to vehicles, e.g., approaching from opposite directions, increasing their sense of the environment. It is essential to highlight that both CAM and CPM messages are broadcast periodically within the proximity of the sender vehicle so that no response message is needed or expected from those who receive the CPMs. Figure 2 presents a simplified diagram of the key states of the periodic message generation flow for an active vehicle (i.e., a vehicle with a running CP service instance in the Facilities layer of its V2X stack).

#### 2.2.2. Technical Details

The original report contained a detailed explanation of how the service should operate, with a precise description of the structure of the CPM messages (with the ASN.1 code appended to the report), which we still consider an instructive read. However, with the newest version of the specification accepted and published in June 2023 [18], some aspects of both the service itself and the CPM structure have been refined, along with the protocol that should support the service. We want to highlight the key aspects of the service and the new CPM format based on this version of CPS.

As a separate entity in the ITS Facilities layer, the CPS may interface with other services and applications to gather information for generating messages and to forward received data for processing. This includes sub-functions like encoding and decoding CPMs, managing the generation frequency, and arranging the decoded CPMs for other entities. The service also supports Multi-channel Operation (MCO) [21], a resource management technique against channel congestion in dense areas.

CPMs are generated periodically with a frequency of 1–10 Hz, managed by the congestion control mechanism of the used access technology. During each generation event, the inclusion of several types of containers and data frames is decided. Timing is also critical since the pieces of information encapsulated in CPMs may be quickly outdated in rapidly moving scenarios. Therefore, the generation of each CPM should not take longer than 50 ms. For the proper apprehension of the messages, each message should be timestamped. The security considerations conform to the trust modeling and certificate usage defined by ETSI, each message is signed, and the certificate includes ITS-Application Identifier and Service Specific Permissions. Apart from the strict timing criteria on the sender side, one other important factor needs to be considered when building an edge application around CP. At the highest generation frequency, a new CPM is emitted every 100 ms. Therefore, this is the time frame where the message must first reach the cloud application, and then the message must be processed to finally be forwarded to all relevant parties and arrive within the time frame. This strictly limits how complex the server-side processing can be before the message becomes obsolete and the QoS decreases.

Figure 3 illustrates the general structure of a CPM. Since the messages are encapsulated with the common ITS PDU header, we will only focus on the actual payload structure, which contains the Management Container and other containers wrapped in a structure called WrappedCpmContainer. The Management Container provides basic information about the originating ITS station, like the reference time, position, and others needed to process the message. Depending on the type of the originating station, an Originating Vehicle Container or an Originating RSU Container is included in the CPM, e.g., the angle and magnitude of the velocity of a vehicle or the map position data of an RSU.

The Sensor Information Container provides data about all the sensors used to detect objects listed in the Perceived Object Container. These include, but are not limited to, a sensor ID, the type of the sensor (e.g., radar, camera, etc.), and parameters about the region the sensor can detect objects in (heading, FOV, range, etc.). It is also possible to express the detection region in terms of simple shapes, like an ellipse or a polygon. Confidence values of a particular region being free of objects (computed by ray tracing) can also be stored in this container.

The Perceived Region Container holds information about each region formerly listed in the Free Space Addendum Container, like the region’s shape and confidence values regarding the object inside the area. One possible application is to expressly state an inhomogeneous free space confidence for a region because of object shadowing. This container can be interpreted without the Sensor Information Container. The Perceived Object Container should first include the number of objects perceived. Due to generation rules, not every object must be listed in each CPM. An ID is assigned to each object, optionally the dimensions, the quality of the perception, and a classification of the object can also be included. Each detected object’s kinematic state is described; this state is interpreted relative to the station position given in the Management Container. Each object may also be linked to the sensor described in the Sensor Information Container that provided the measurement data using the sensor ID. Since not all detected objects are present in every message, stations running the service should be able to keep track of the relevant ones.

### 2.3. V2X Services Support in 3GPP Infrastructures

#### 2.3.1. 5G Architecture

To use NR-V2X for advanced applications, 5G must provide architectural support. ETSI TS 123 287 defines such enhancements developed by 3GPP [22]. This means that, on one hand, the architecture must support new types of user equipment (UE) that use either of the two NR-V2X interfaces, PC5 or Uu, and the interfaces used by V2X applications within the network. On the other hand, high-level features must also be defined that guarantee the functioning and the QoS of V2X applications. The specification contains models of roaming and non-roaming scenarios, as well as a description of the functional entities of the network with the extended functionality related to V2X scenarios. A logical design choice is that RSUs are not separate entities but only implementation options; V2X application logic can be implemented on any UE or gNodeB equipment, thus realizing an RSU.

The section on high-level functionalities starts with listing the options for UE authorization and how the parameters of the security mechanisms should be distributed. This is also a key factor, as V2X systems will be an appealing target for people with malicious intent. When using the PC5 interface, the communication itself can be broadcast in the case of LTE PC5 and can be broadcast, unicast, or groupcast when using NR PC5. IP-based and non-IP communication is also supported. However, only IPv6 is used in the case of the former. Hence, IP addresses are allocated via SLAAC or link-local addresses. Over the Uu interface, only unicast communication is supported; UEs equipped with both interfaces can choose which one to use in certain situations. UE messages are routed towards a V2X Application Server if the target is yet to be decided or towards a UE with an existing unicast routing when using the Uu interface. The address(es) of V2X Application Server(s) may be pre-configured on the UE or provisioned over the N1 interface (see Figure 3).

A new paradigm gaining more attention in scenarios not only related to V2X but also IoT, in general, is called Multi-access Edge Computing (MEC) [23]. The idea is to bring computational resources to the network’s edge, as close to the end users as possible, resulting in much lower latencies and reduced load on the core network. The reference architecture is standardized by ETSI [24], which enables the on-demand instantiation of MEC applications on nearby MEC servers that can also subscribe to services hosted by the platform. The platform also provides support for resource and mobility management; therefore, the app instances can migrate from host to host as the vehicle moves. Although it is clear that this is a challenging task, MEC is a potential asset in realizing Day 2 V2X applications [25,26]. Needless to say, MEC-assisted V2X would be impossible if 3GPP did not work on MEC support in the 5G core in parallel to ETSI developments [27].

#### 2.3.2. ETSI V2X Information Service

Regarding V2X, the devices communicating will be manufactured by multiple vendors and connected to numerous networks using various access interfaces. To facilitate interoperability in such an environment, ETSI has specified a new service, namely the V2X Information Service (VIS), as well as the data flows, data types, and the API required for accessing the service [28]. The document considers scenarios that combine single or multi-operator situations in terms of OEMs and Mobile Network Operators (MNOs) providing certain V2X services. Figure 4 illustrates how VIS enables “horizontal communication” between MEC services, thus interconnecting devices in different MNOs with lower end-to-end latency. It also implies the need for mobile network-level cooperation among other operators to ensure user service continuity. The VIS API further increases the accessibility of MEC in automotive scenarios involving a wide range of car manufacturers, device OEMs, MEC vendors, mobile operators, etc.

MEC VIS includes the following functionalities:Gathering of PC5 V2X-related information from the 3GPP network (authorized UEs, subscription info, configuration parameters);Exposure of this information to MEC apps;Enablement of secure communication between MEC apps and the logical functions in the core network;Enablement of secure communication between MEC apps in different MEC systems;Possibly gather and process information available in other MEC APIs to predict RAN congestion and notify UEs.

Relevant information to MEC apps is exposed via the Mp1 interface. To enable access to the core network’s V2X Control Function for VIS without much overhead (e.g., to obtain user subscription data), the group specification suggests that the 3GPP V2X Application Server could be deployed in the MEC system as a MEC app alongside VIS. The VIS could also be essential in the discovery and connection establishment of inter-MEC system applications. Similarly, services might also be exposed between different MEC systems so that a service appears in the service registry of the local MEC host (from the UE’s perspective). VIS is also envisioned to provide a journey-specific QoS prediction in the highly mobile environment where rapid changes in the quality of the radio connections are inevitable.

### 2.4. Collective Perception Use Cases

The widespread use of CP-based applications would primarily impact traffic safety. In urban and rural scenarios, important data could be exchanged among vehicles, which could then be displayed for the human driver or trigger a direct intervention in the vehicle’s handling as a form of a driver-assistance system. Although CP has much potential, these use cases are mainly conceptual without real-world implementations due to the lack of a final and comprehensive standard.

Some examples mentioned in [17] are cooperative overtaking and cooperative merging. Regarding cooperative overtaking, CP could help assess the situation and determine whether the maneuver can be safely executed when visibility is obscured (e.g., behind a cargo truck). Simple visual messages could be displayed to the driver (lane free/not free), or even a live video feed could be broadcast in some cases. Cooperative merging on highways utilizing the shared environmental model could result in a safer and optimized traffic flow, even with non-connected cars. The advanced version of this application could even share vehicles’ intentions and trajectory information, which is an excellent asset for sophisticated algorithms controlling automated cars. However, unlike cooperative overtaking, this technology is hard to imagine working effectively with human drivers. It would be challenging to make them abide by the recommendations to slow down or change lanes appearing on their displays.

CP could also enhance the protection of Vulnerable Road Users (VRUs). Passive protection could mean that a system deployed, e.g., at a pedestrian crossing, broadcasts information about pedestrians obscured to some vehicles (therefore, passive means that the pedestrians themselves do not broadcast messages). While in the form of active protection, VRUs could use their phones or devices embedded into the vehicle to warn other vehicles about their presence [29]. It is essential to mention that a custom message format for VRU protection called VRU Awareness Message (VAM) is already under development by ETSI [30].

A third study cites four other use cases for CP in hybrid scenarios, where V2X capable co-exist with plain vehicles: Emergency Electronic Brake Light (EEBL), Left Turn Assist (LTA), Intersection Movement Assist (IMA), and Blind Spot Warning (BSW) [31]. EEBL can notify drivers of a sudden breaking, reducing the likelihood of a rear-end collision. IMA and LTA could minimize the possibility of collisions near intersections by warning drivers of incoming vehicles, LTA specifically by warning of oncoming traffic where a left turn is to be made. Finally, BSW, an already well-known safety feature, could be extended by including CPMs as an addition to local sensors.

### 2.5. MEC-Assisted Use Cases

Effectively using a Collective Perception-based application should rely heavily on both CPS and a local database of perceived objects. Even MEC could have a significant role in implementing sophisticated services, such as Sensor and State Map (SSM) Sharing, an advanced implementation of said database described in [32], or in incorporating V2X-integrated multi-layered HD maps [33]. As a review of possible use cases for MEC-supported C-V2X/NR V2X applications using CP, two groups of use cases mentioned in a 3GPP study will be introduced briefly [34].

#### 2.5.1. Extended Sensors

This group consists of use cases related to most kinds of traffic participants (not capable of communicating, V2X capable, and V2X capable with other sensors) implementing the basic ideas of CP. These include vehicles, RSUs, street cameras connected to the Internet, or even pedestrians carrying smartphones. Falling into the most equipped group (V2X + sensors), they share the preprocessed sensory data via V2X to alert each other. This data can also be sent to feed instances of SSM or an HD map rendering application deployed on MEC servers, which in turn ensures lower latency and offers more computational resources, e.g., for further data processing. Video data sharing for automated Driving (VaD) might be an improved version of classical CP utilizing MEC servers [32]. Vehicles using VaD relay the raw HD video stream instead of preprocessing the available data, and MEC would be suitable to help in transmission control tasks to provide on-demand access to these streams. These services, by nature, imply the idea of handling latency-critical and computationally heavy issues locally, near the users, making the use case group a candidate for using MEC. Cooperation of these services with other applications and special routines implemented on a local MEC host could also result in a more effective control system for automated vehicles in the future.

One of the most significant problems of realizing such applications feeding from multiple data sources is data fusion. There are two main challenges for the use cases above. The first is the exploitation of the multisensor capabilities together with the creation and continuous maintenance of a much broader view/model of the environment, while the second one is tracking the detected objects as they move around. This field holds some interesting yet unanswered questions concerning scientists and industry organizations. As an entity envisioned to be responsible for a limited area, a MEC server could host an application that tracks pedestrian movement based on object-level fusion [35] and provide additional functionality, like sending VRU warning messages for vehicles based on predictions of dangerous situations. Aside from MEC-based use cases, object fusion is also a key enabler for automated driving scenarios, and, as such, it interests different stakeholders experimenting with field tests [36] and simulated software environments alike [37].

#### 2.5.2. Advanced Driving

Instead of building upon the basic features of CP, advanced driving collects higher-level applications (most of them handling automated vehicles), enabling more optimized traffic and a safer experience for drivers. Not only sensory data is transmitted, but also intentions for various maneuvers and pre-calculated trajectories. The following use cases make up this group [32]:Cooperative Collision Avoidance (CoCA)Information sharing for limited automated drivingInformation sharing for fully automated drivingEmergency Trajectory Alignment (EtrA)Intersection Safety Information Provisioning for Urban DrivingCooperative lane change (CLC) of automated vehicles3D video composition of V2X scenario

CoCA is meant to coordinate fully automated vehicles by combining extracted sensory data with the vehicle’s intentions and sending them via the usual V2X messages. The complexity of information sharing depends on the level of automation (limited/full). Limited automation usage might require the remote intervention of a human operator (using remote driving, which is also an interesting 5G V2X use case by itself), whereas, in fully automated systems, the vehicles must handle every situation using their implemented logic. Therefore, it can be concluded that partial knowledge of the surroundings and shared state information might be adequate for limited automation, full automation also requires highly processed sensor data and accurate trajectory estimates.

EtrA is used to rapidly calculate emergency trajectories if an unexpected obstacle appears and then broadcast the intentions for safety maneuvers, thus building upon CP features. With CLC, automated vehicles merging onto highways and changing lanes could choose their position more safely and effectively using shared intentions and trajectories. 3D video composition closely resembles SSM. The application server collects video feeds streamed by vehicles (or other cameras) and renders a to-scale and accurate 3D video of a certain region, which can be used for safety solutions or even to reconstruct details of an accident for forensics [32].

Many of the above use cases could benefit from locally deployed MEC servers. In the case of CoCA or EtrA, the user devices could offload demanding subroutines, and a centralized trajectory calculation could optimize emergencies for all involved parties. 3D video composition on this scale is probably also unimaginable without high-capacity servers. However, this partial or complete control or management logic centralization still creates additional latency (though not as much as in the case of a remote data center), potentially compromising the purpose of the above safety applications. In the end, it is a tradeoff the engineers planning and deploying such systems will have to consider.

## 3. Related Works

The most inspiring prototype for our modeling and simulation work was a complex initiative to reach NR-V2X-based autonomous platooning assisted by a CP-based MEC service [33]. The system took advantage of 5G’s support for the MEC architecture to host an HD 3D map service boosted by custom deep-learning and swarm intelligence-based algorithms. In addition to the works introduced in our original conference paper related to the extension of CP-based scenarios with MEC [20,38,39], we would like to introduce two more studies that impacted our approach to the topic.

It is now widely recognized that MEC can be a powerful solution for ever more demanding low-latency and high-reliability/high-bandwidth use cases. In [40], the authors proposed a four-layer approach to the general architectural design of MEC supporting C-V2X scenarios. This design is based on an earlier version of the reference architecture published by ETSI [41], but it emphasizes the heterogeneity of participants of V2X scenarios. The authors aim to improve the capability of MEC to meet the requirements of a broader range of end-to-end services in multi-access and multi-vendor scenarios. To achieve this, a more refined structure of vertical layers is presented, followed by the definition of new service APIs for implementing C-V2X services utilizing MEC and two examples of possible implementations using the proposed architecture.

Most of the elements have a counterpart in the reference architecture, like the general platform, along with the different app instantiations. Still, this approach makes the deployed system look more straightforward, as each of the various participants, like the cloud operators, roadside infrastructural elements, and the user vehicles, are represented by their layer. This way, it is visible how the reference architecture does not clarify the method of interaction between devices of different manufacturers, network operators, etc. An entity called V2X Server is also abstracted from the general hardware pool. Its purpose is mainly parallel to the MEC platform in the reference architecture; it is responsible for managing registered applications and services.

Though the reference design specification has been updated since 2019 with new features, the core of the architecture is identical. However, the idea of a V2X Server as a somehow separated entity reflects the ideas of VIS and the 3GPP V2X Application Server described in Section 2.3. The proposed architecture was of much use when designing the different elements of the simulation models seen in Figure 5 and our scenario described later in Section 4.

One of the most significant problems concerning MEC applications is the application and service placement to support the highly mobile V2X use cases. The migration of these services between the edge nodes requires some of the resources of each node to be allocated for this purpose only, which, if done incorrectly, could seriously impact the performance of the edge servers, resulting in higher end-to-end latencies for the V2X applications. In [42], a proposal is made for how complex V2X applications should be decomposed into the pre-defined V2X essential services and a linear programming model for optimal application placement based on delay and resource requirements. The basic services in this context mean, e.g., those standardized by ETSI, like the CA and the Decentralized Environmental Notification (DEN) services, though this principle should apply to future Day2 and Day3+ services as well, like Collective Perception and VRU Awareness. The authors tested the presented heuristic algorithm under realistic simulation scenarios with different traffic conditions, and the results showed that the approach could guarantee an acceptable quality of service. Even though our research has covered a scenario with a single MEC host so far, the significance of this research will undoubtedly be helpful to us in the future as our proposed framework evolves.

We aim to present an appropriate simulation environment for the large-scale evaluation of V2X applications using CP with MEC support by integrating and extending the available open-source libraries of the Artery/OMNeT++ ecosystem at present. Table 2 presents a comprehensive collection and comparison of the existing initiatives for simulation model implementations.

To our best knowledge, no such framework has been proposed to this day that simultaneously supports advanced ITS-G5 Facilities (like CP) and 5G protocol stacks. However, many other attempts have emphasized the justifiability of MEC-based systems in V2X use cases. These include model or framework proposals and sterling simulation results, further proving that many parties are interested in researching this topic. Despite the promising capabilities of simulating accurate network behavior, the one severe weakness of existing solutions is that they focus on one or, at best, two of the following critical aspects of a complete, realistic MEC-assisted V2X simulation model: 5G cellular network (incl. MEC), environmental model (traffic simulation + sensor suite), and the simulation of the ITS protocol stack. That means the missing part must usually be generated based on some available statistical data or over-simplified to completely hide the given aspect’s representation. One outstanding exception is the framework CrowNet used for a crowd-sensing strategies study [56]. This integrated framework consists of several open-source projects that also have an important role in our solution (see later in Section 4). The need for a slightly different approach is because CrowNet has a greater focus on integrating massive pedestrian mobility into the complete ITS simulation environment, whereas the subject of our study is the integration of collective perception and edge computing from the automotive use cases perspective. An interesting extension in CrowNet is the addition of control plane features to device-to-device (D2D) communications via the PC5 interface, something which is not yet present in our 5G model. However, it is on our roadmap to implement the control plane to be able to simulate the communication over the PC5 interface according to NR-V2X specifications. 

With the attempt to fix the mentioned shortcomings of the current state-of-the-art, the motivation behind this work is to create an integrated framework capable of simulating an accurate environmental model with a dynamically pluggable, diverse sensor suite to feed our CP model [58] and, from a network perspective, the traditional ITS stack as well as the 5G cellular connectivity with the addition of MEC infrastructure. Such an environment can be utilized for many sub-fields of MEC-assisted V2X research. These include, but are not limited to, developing and testing a wide range of V2X safety applications that are based on CP, conducting endurance or load tests from a network perspective (like channel congestion), or the study and optimization of the size and resource pool of MEC nodes in highly connected automotive scenarios, which can help develop automated algorithms for dynamically creating the necessary network slices based on traffic load. The modular nature of the framework that is inherited from the OMNeT++/INET-based ecosystem (see Section 4.1.2) further increases the possibility of putting more focus on a selected part of the model (e.g., load balancing, network latency/properties, edge computing resource optimization, etc.) when running simulations, by plugging in simpler or more refined modules, depending on the research needs.

## 4. Modeling and Simulation

### 4.1. The Used Simulation Libraries

The most available assets are simulation tools for rapidly testing ideas and implementing POC scenarios in the V2X domain. These tools must accurately represent and calculate nearly all aspects of a subject’s real-world behavior, making it practically impossible to obtain accurate results for all network traffic, physical participants, etc. That is why the current practice for researchers is to rely on multiple libraries and standalone frameworks, each focusing on a single part of the target environment (e.g., network stacks, traffic simulation) instead of a massive built-for-all simulator. In this section, the structure of our integrated framework will be described, and the used libraries will be briefly introduced.

#### 4.1.1. Vanetza

Vanetza is an open-source project presenting a C++ implementation of the ETSI ITS-G5 stack [59]. Primarily, it connects the application and radio access layers, featuring each layer and specific entities in the protocol stack. It implements various ITS protocols and facilities, like Basic Transport Protocol (BTP), Dynamic Congestion Control (DCC), and GeoNetworking, but one of the strongest features is the support for proprietary message formats written in the ASN.1 language.

#### 4.1.2. Artery

Artery is a fully fledged framework for simulating V2X applications [60] that is built upon OMNeT++, a widely used discrete event simulator. Originally, it was created as an extension for Veins [61], another similar framework. Over the years, it has become a standalone environment, mainly to implement European standards (instead of the ones defined by American organizations) and to enhance the simulation capabilities by allowing multiple applications to run per vehicle, something Veins was incapable of. What makes Artery unique and powerful in this domain is its Middleware functionality, serving as an additional layer of software between the application instances and lower layers (like network or physical) provided by, e.g., Vanetza. Essentially, the Middleware offers useful interfaces in both directions while managing application lifecycles and actively taking a role in transmitting messages. Furthermore, Artery’s Middleware instantiates the whole stack in each vehicle in the simulation to recreate real-world behavior as precisely as possible.

Besides libraries providing the C-ITS stack, like Vanetza, Artery also depends on others that are responsible for the access and physical layers. Two alternatives are available: these layers are already implemented by Veins, but INET [62] can also be used, which is a standalone library for simulating mobile, wireless, or wired networks. Even though INET provides basic mobility support, for the precise simulation of vehicle movement the extensive traffic simulator, SUMO (Simulation of Urban Mobility) is used. Much like Artery, SUMO is a microscopic simulator, meaning that an individual model is parametrized, created, and maintained for all the nodes with programmed trajectories.

Although Day 1 V2X applications (e.g., Collective Awareness Basic Service, Decentralized Environment Notification Basic Service) are pre-built into Artery, our research concerns Collective Perception, therefore our own CP model had to be integrated, updated, and used [58].

#### 4.1.3. Simu5G

Simu5G [63] is responsible for integrating 5G network elements into the simulations. It is based on OMNeT++, too, and implements 5G core functionality, NR access technology (Uu interface), and other network elements, like the gNodeB and functions of MEC infrastructure. It also depends on INET, meaning that nodes using 5G/4G/3G or other access technologies can fully cooperate within the simulations.

### 4.2. The Integrated Simulation Framework

Using different frameworks and libraries to create a joint environment often results in compatibility issues. Simu5G, the most recently created among the listed ones, was the ‘bottleneck’ during the process. The latest version of Simu5G requires INET v4.4.0, and we used OMNeT++ 6.0.1, which works with both INET and Simu5G. The exact version of Simu5G was v1.2.0. However, there were constant updates since that particular version was tagged with no changes in version numbering, so we decided to always use the latest developments from the original GitHub repository [64].

#### 4.2.1. Integration of Artery and Simu5G

In [65], Artery’s developers suggested using CMake to manage the build system when additional libraries are inserted. Thus, the integration of Simu5G into Artery seemed manageable at first sight. CMake code already shipped with the framework building external libraries, like SimuLTE (similar to Simu5G, implementing 4G elements), was easy to comprehend and reuse for our purposes. However, we faced one problem when linking proprietary code that depended on Simu5G to Artery’s pre-built features. Specifically, linker errors were thrown when extending the build system using the add_artery_feature macro. The source of the problem was the linker not being able to find the Simu5G shared library (after a successful build) due to naming issues, possibly because of a glitch in OMNet++ CMake macros or the provided *opp_cmake.py* script. Rooting out this naming issue would be the desired and more general solution for the problem, but for the time being, we opted for a simpler workaround. We created a completely new macro to hard-code the Simu5G include directories for new features that depend on the library, a solution that finally resulted in successful builds with code from Artery and Simu5G working together.

Artery does not yet support OMNeT++ 6.0 (which has different submodule handling than 5.x versions), which is a requirement for Simu5G. Thus, we had to adjust some parts of the TraCI API. In the newer versions of OMNeT++, submodule lifecycle management is rather stringent, which affects the dynamic insertion and deletion of submodules. Since it is an essential part of Artery to instantiate car modules based on the SUMO simulation, this issue had to be dealt with. Fortunately, the developers of another similar project combining multiple simulation frameworks, namely CrowNet (https://crownet.org/, accessed on 10 July 2023) have already ported many parts of Artery to OMNeT++ 6.0, and we could reuse some of their solutions.

As a result of this integration step, the available assets to our simulation model have been successfully extended with all the 5G radio and core elements, along with the ETSI-compatible MEC implementation provided by Simu5G. These elements had to be switched both in the car model and the existing world model to replace the LTE radio equipment and EPC core functionality. Additionally, a MEC node module has been connected to the simulated network setup to host applications for the cars.

#### 4.2.2. Our Extensions to Simu5G

Understanding Simu5G’s implementation of the MEC platform and implementing the client- and edge-side applications was intuitive, thanks to the many example use cases. Based on these examples, our idea was to create a MEC application as simple as possible but one that fits in the vehicular scenario and is modular enough to make future changes straightforward. In the meantime, there have also been updates in the implementation of the MEC application interfaces since the publication of our original paper. Therefore, all the relevant changes had to be mirrored in our own MEC app to conform to the API changes.

As a first step for our implementation, the CPMs had to be delivered to the MEC host to simulate processing efforts and a higher-level application feeding off the received data. The framework allows instantiating supported MEC applications at will; therefore, we extended the available application pool with a CP service application capable of receiving data from one sender vehicle. Each vehicle instantiates an application on demand.

The architecture of this setup can be seen in Figure 5, and a simplified, class diagram-like representation of the implemented modules in Figure 6 (with our implementation highlighted with a slight fill in the boxes). Each application instance running on the MEC host can only access the message stream of one individual vehicle, meaning that any aggregation of the incoming data, e.g., for object fusion or other safety-related applications, cannot be done without either connecting the app instances or introducing another entity within the MEC host. The first approach would exponentially increase the applied connections between these submodules, and it would also break modularity and the single responsibility principle for these app instance modules (with the relevant responsibility being the reception of messages and relaying them for further processing). Therefore, the second approach seems to be more promising, especially if we consider that the ETSI MEC framework supports the hosting of services, like Location Services [66], commonly available for each application instance running on the host.

We also had to generate traffic from the MEC app instances back toward the vehicles to obtain results for our measurements. Typically, the service mentioned above could feed the vehicles, e.g., with fused data. For a simple model of this, we utilized the fact that the app instances in the MEC host have an established connection to their counterparts on each vehicle, and as such, we simulate the processing time of a higher-level service, and we send back metadata of the exact CPM that triggered the processes in the edge platform, so the server application can confirm that the message was received and digested. Our model for these server-side processes is further described in Section 4.3.

It is also relevant to mention how the CPMs are represented/digested by the edge app instances and how further metadata are obtained. As already discussed in Section 4.1.2, we reused and updated an already existing Collective Perception model [58], including the ASN.1 representation of the CPM. Creating and handling ASN.1 message formats is straightforward, thanks to wrapper classes and other helper techniques provided by Vanetza. The available methods to serialize/deserialize messages also result in convenient transferring of the messages between Vanetza and INET frameworks. As a reminder, Vanetza provides implementation to handle V2X messages, like CPM, whereas our model’s cellular and other network components are based on INET implementation; therefore, data conversion between the two frameworks is inevitable. That means that the edge application instances can digest the same CPM that was created by the CP service running in the car’s V2X stack. This includes all the fixed and optional components of the message described in Section 2.2.2. The most relevant message container for us was the Perceived Object Container, which is directly involved in the scenarios where the processing time is dynamically calculated, as described in Section 4.3.2. As the whole message structure can be rebuilt during deserialization, the number of perceived objects is easily obtained in the edge application. Because the nature of the CP service does not require the identification of the exact messages, we had to add an extra field before each CPM for sequence numbering, managed by the client cars. This sequence number is the only metadata that is sent back to the client after processing the message to keep additional network load to a minimum. With this metadata, the client can independently calculate the accurate response time for the messages. Furthermore, with each message identified, the calculation of other statistics, like packet loss, could also be easier.

In summary, our model, for now, consists of vehicles propagating CPMs over IP/UDP to an application instance hosted on the local MEC server responsible for the dissection and processing of the messages. This is a minimal working network setup, and the model is planned to be further extended later with the implementation of higher-level applications (e.g., for an object fusion service or different use cases described in Section 2.5) within the MEC host’s service registry that is accessible for the app instances via REST API [67], where the now simplified processes could be replaced with actual extended functionality.

### 4.3. Simulation Scenarios, Parameters, and Initial Results

The idea behind the whole scenario was to test the integrated framework with simple but successful simulation runs so we could set up a proof-of-concept verifying how MEC usage could positively impact CP-based use cases in the long run. The test simulations were run in a grid-based urban scenario. We have changed the map since the original article so that the measurements can focus more on the performance of the MEC infrastructure under heavy load rather than the realistic traffic. In the scenario under investigation (see the networking architecture in Figure 5), vehicles move inside a four-by-four grid of 100 m-long road segments displayed in Figure 7. Multiple devices are deployed at the center of the grid: a gNodeB with a MEC server.

The cars start transmitting the CPMs to the MEC server over the NR-Uu link using a 2.1 GHz carrier. The receiving application instances simulate the endpoint for a complex service, like SSMS, running on the server and being actively responsible for the region. To achieve this, V2X-capable nodes in the simulation had to be equipped with a new CPS application module that relays the CPMs originating from the Artery Middleware. The integrated framework can successfully simulate the network traffic sent over the NR-Uu interfaces using Artery-based Car model extensions, the MEC applications instantiated without any problems, and the reception of CP information (originated from the vehicles’ Middleware CP service) at the appropriate MEC modules.

The total number of vehicles in the simulation constantly increases to put a growing load on the edge cloud infrastructure (see Figure 8). With the network architecture and the traffic generation set, we could start testing the behavior of the MEC server by altering some of the simulation parameters. We experimented with some of the options mentioned by the authors in their thorough description of the MEC capabilities of Simu5G in [57]. Our experiments covered different cases of physical resources available on the server and the resources required by the app instances. The model works with MIPS regarding the CPU computational power (both server capacity and app requirements are given in MIPS). Additionally, the available and required memory (RAM) and disk capacity can also be set for each application instance (not on a per-instance basis, the values are common for all instances). Our focus was on the computational toll and, as a direct consequence, the processing time of the incoming CPMs, so the imagined edge-side CP service’s response time was modeled using the built-in method calculateProcessingTime, assuming a fair sharing model for the resource distribution among applications. We define the total response time shown below in the graphs and Table 3 as seen in Equation (1), i.e., the UL delay for sending the CPM to the MEC host, the processing time of the CPM, and the DL delay of the response message. The simulations can be further divided into the following two categories: one with a set of fixed CPU requirement parameters and one where the simulated processing time is based on information contained by the CPMs.
resp = d_UL_ + t_process_ + d_DL_(1)

#### 4.3.1. Simulations with Fixed CPU Requirements

In this subset of simulation scenarios, the MEC host has the same CPU capacity in all runs, but the client applications have a different CPU requirement value between each run. All the application instances have the same constant value for each function call within the same run. Table 3 summarizes the key parameters (with some results, too) we changed to test whether the integrated simulator responds accordingly to the increased workload. We collected the response times for each sent CPM in each vehicle, our primary focus. Still, we also recorded the end-to-end network latency from the cars to the MEC app instances, a significant portion of the response time [57]. Results about the average end-to-end latency, which are also seen in Figure 9, show that the network latency (calculated between source vehicles and corresponding processing MEC entities) experienced by the majority of the nodes in this particular scenario is around 12–15 ms with even the most extraordinary outlier below 50 ms. These outliers occurred due to some messages taking about 1 s to arrive, which rarely happened. The increases in this value did not seem to correlate with the increase in vehicle density over time.

However, noticeable changes appear in the response times defined in Equation (1). Table 3 only summarizes the average value calculated for each entire simulation run, i.e., taking the average of each node’s average response time, which itself highlights the increasing tendency. This tendency is presented in more detail in Figure 10. The boxplot diagrams show the distribution of the experienced response times of the nodes during simulation runs with different CPU requirement values. Notice that not only the median values are increasing but also the interquartile range, as well as the range of the whole data set. This might prove that either the degrading quality of service might also mean that there is increasing stochasticity in addition to the anticipated increases in response time (since the distribution of the values is much wider) or that the model simulating the computational load is simplistic and is a limiting factor. Regardless, these results show that the initial model is working and ready for further enhancements drawn from the adequate conclusions.

The results showing data measured in individual car instances are especially insightful when an application instance requires 300 MIPS or more. In these cases, the performance drastically decreases even with this relatively small number of vehicles (approx. 80 vehicles by 30 s of simulation time), and the increasing trend can be easily observed regarding response times. Histograms drawn using the average and the median of the measured response times for each vehicle, shown in Figure 11, are excellent examples of this performance drop. Each pair of columns represents the measured data of one car module, with the first modules illustrated on the left side of the graph and the modules that entered the last on the right (i.e., from the least loaded environment to the maximum load in the evaluated MEC architecture).

Looking at a selected node that entered the simulation quite early (at 2 s, specifically), a correlation can be seen between the graph drawn from the measured response times in Figure 12 and the increase in vehicle density (Figure 8). This, of course, is not surprising at all. Still, it is a relevant observation, as it shows that the framework could handle scenarios with actual services and algorithms implemented on the MEC host to get real-world-like results with appropriately selected parameter sets of future research efforts in the domain.

The reason for choosing the applied capability and CPU requirement parameters is quite simple. As for now, we did not aim to identify the properties of a real implementation of a complex V2X service but to provide a glimpse of the potential and limitations, a proof-of-concept for the operability of the integrated simulator. Although Figure 9, Figure 11 and Figure 12 show graphs based on measurements where the required CPU capacity of the MEC apps was 300 MIPS, as the changes were most visible in this case, some interesting conclusions can also be drawn from the other cases. The increase can also be observed in the case of 50 and 100 MIPS, but a stable value of around 40 ms is quickly reached.

The value of 300 MIPS was much more demanding, as the vehicles entering the simulation early could experience worse than a threefold increase in response time throughout the simulation (see Figure 12). Still, the maximum values were around 80–85 ms, which is not ideal for V2X services like CP but is under the aforementioned 100 ms barrier (as in Section 2.2.2) that is considered necessary for such scenarios. Setting the requirements to 500 MIPS, however, breaks this barrier. Directly addressing the problem mentioned in Section 2.2.2, this means that the maximum acceptable complexity for an algorithm processing each message arriving at the edge application is somewhere between 300–500 MIPS for this particular model. Even though the average response time (also seen in Table 3) is just around 100 ms, some averages even reached 120 ms, meaning that CPMs generated with a frequency of 10 Hz would be obsolete when the response arrives. This observation might be useful when designing such edge services or algorithms to be able to balance the problem of required/available computational power of the MEC apps/server depending on the vehicle density. However, we have already noted that a more sophisticated parameter setup is needed for such evaluations.

#### 4.3.2. Simulations with Dynamic CPU Requirements

For the following subset of scenarios, we tried to implement a more realistic behavior that is a more sophisticated model of the CP edge service performing object fusion, described in Section 4.2. To achieve this, the processing time for every CPM changes dynamically depending on the number of perceived objects whose information is encapsulated in the relevant container of the message body. This approach can affect MEC performance in two ways. As the number of vehicles increases in the simulation, the MEC host must provide service to more and more clients, which is the same as before. However, with all the clients having different experiences with their environment, their specific computational requirements are expected to vary throughout the simulation, resulting in more apparent fluctuations in the recorded response times.

To test the actual impact on the edge server, there are three obvious parameters to experiment with: (1) vehicle density within the simulation, (2) parameters of the sensors detecting these vehicles, and (3) how the increased number of perceived objects affects the required CPU performance, i.e., the equation describing this relation in our model. As for the vehicle density, we generated much denser traffic flows for the same map introduced in Section 4.3. The number of cars simultaneously in the simulation was still capped at around 80 vehicles, but this number was reached more and more quickly as we created denser traffic. The cars in the simulation were equipped with front radars, with a detection range of 180m and a FOV of 120°. The range and the FOV were arbitrarily chosen so that as many vehicles could be captured as possible. However, during the adjustments to these parameters, we encountered a problem regarding the number of perceived objects. The packet size could exceed the maximum data unit size of a GeoNetworking packet [68] if there are too many detected objects, causing an error in the simulation. Therefore, as a temporary solution, we had to limit the number of objects embedded in the CPMs to keep the simulations running. Normally, this would not be a problem because CPM dissemination and data inclusion rules [18] state that the total number of perceived objects (which is a mandatory field in the Perceived Object Container) and the number of additional perception data elements (i.e., the additional data of some detected objects) do not necessarily have to be equal. However, since our model does not include object tracking yet, our V2X stack was packing object data information into the CPM without any prioritization, eventually exceeding the data unit size limit. Implementing data inclusion rules specified by the standard or any proprietary solution would eliminate this problem. Still, for the sake of simplicity, the maximum number was maximized at 25 objects for these simulations. Lastly, there were no hidden limitations when constructing the dynamic model of the CPU requirements. To remain consistent with the previous set of scenarios and to counter the limitations previously discussed regarding the number of detections in a CPM, we adjusted the parameters of a polynomial so that the required CPU MIPS would reach 500–600 around 20 perceived objects (see Equation (2) below) to stay conforming to the highest possible MIPS values within the range of acceptable response time (see Section 2.2.2) obtained in the fixed parameter scenarios. The applications on the MEC node obtain the required number of CPU instructions, and this result is used to calculate the necessary processing time.
y = −0.1353x^3^ + 4.07x^2^ + 2.6333x + 2(2)

In this dynamic approach, the processing and response times are directly affected by the number of perceived objects. Therefore, it is best to first analyze the results in this regard. Figure 13 shows that denser traffic results in the expected increase in the average number of total detections aggregated in the MEC host. Comparing the two subplots (a) and (b), it can be concluded that by making the traffic three times denser, the average of total detections doubled by the end of the simulation. The trends in the increasing number of detections are also similar to those regarding the number of simulated cars, further reassuring that the measurements are consistent (see respective graphs in Figure 8). Note that the incoming data to the edge application instances, i.e., the numbers of perceived objects, were simply summed up since this model does not contain any object tracking or fusion yet. It is expected that using, e.g., object fusion would result in similar graphs with numbers closer to the actual number of cars present in the simulation.

With this many overall detections, we expected that denser traffic would also result in the degrading performance of the MEC host and the increase in the response times experienced by the cars, as it did by increasing the MIPS requirements in the scenarios with fixed parameters. To see how many cars had the most demanding requirements, we analyzed, on average, how many cars had more than 20 detections throughout the simulations. From what is observed in Figure 14, and also based on what was seen in the fixed 500 MIPS case, a sound expectation might be that the increased number of cars with “many” detections would cause higher response times and variance in the results for denser traffic scenarios. The obvious increasing trend in the case of the densest traffic is only one clue. The difference between the maxima of such cars in the simulation runs measured with the lightest and the densest traffic is also significant. With the least dense traffic, the maximum is 8, i.e., one-tenth of the total number of cars, whereas, in the case of the densest traffic, it was 35, almost half of the simulated vehicles.

Looking at Figure 15, it can be observed that, although there is a marginal increase in the median of the average response times, the variance barely changes and the outliers also tend to be smaller. It seems that, despite our expectations and our previous results in Figure 10, the simulation runs with denser traffic were more stable in terms of performance. The reason behind the fact that a similar increase to what was present in the first type of scenario cannot be observed now might be that even if at some points during the simulations almost half of the cars had heavy demand, the number of such cars was constantly fluctuating. Therefore, at no single point in time did all the cars require that many resources (contrary to the first type of scenarios), and the MEC host was able to serve requests without a significant drop in performance. As for the reduction in the variance of the observed response times, we deduce that the outlier values gathered most of the extra latency at the network level, which again might occur due to anomalies in the OMNeT++ framework’s event system. Further investigation of both the resource allocation methods in the MEC implementation and the event management of OMNeT++ might give a better insight into the exact reasons for the obtained results. It is, nonetheless, a great experience that the integrated simulation framework could be successfully used in testing both MEC performance and advanced Collective Perception-based V2X services in various scenarios.

## 5. Conclusions

The deployment and testing of V2X applications and novel algorithms are, for the most part, easier, faster, and cheaper for researchers using simulation frameworks, among which Artery plays an important role as one of the most practical tools. Extending the framework with 5G elements implemented in Simu5G, especially MEC, enables the study of CP-based advanced applications on a whole new level. The integrated simulation environment we present will significantly ease the further study of applications that belong to the use case groups listed in Section 2.5 and many more. According to the limitations discussed in Section 3, further advancements in the implementation (e.g., NR-V2X PC5 implementation, CP MEC service for object fusion) and future scenarios might also include comparing the NR-Uu interface-based operation with ITS-G5 or NR-PC5 access types and analyzing collective perception performance with different sensor fusion schemes over various MEC optimization possibilities. With the inclusion of a complete NR-V2X implementation, the simulator could also be used to test different resource acquisition methods [69], potentially investigating the role of MEC in the process. V2X messaging alternatives over the Uu interface can also be tested, for example, how AMQP/MQTT or other message queuing solutions could perform for this purpose. It would also be interesting to see whether standalone 5G (Uu+PC5) can satisfy all the requirements or if hybrid-RAT-based systems might be needed where ITS-G5 and cellular schemes work together. Additionally, further refinement of the model/equation of the necessary dynamic processing time requirement based on real-world data or using a more sophisticated statistical apparatus will also be an interesting challenge for our future work. We truly wish that our contribution will benefit the research community interested in MEC-based vehicular applications, either by serving as a ready-to-use sandbox for testing algorithms for CP-based V2X applications utilizing MEC or by providing a stable platform for even more complex scenarios hosting advanced V2X services.

## Figures and Tables

**Figure 1 sensors-23-07968-f001:**
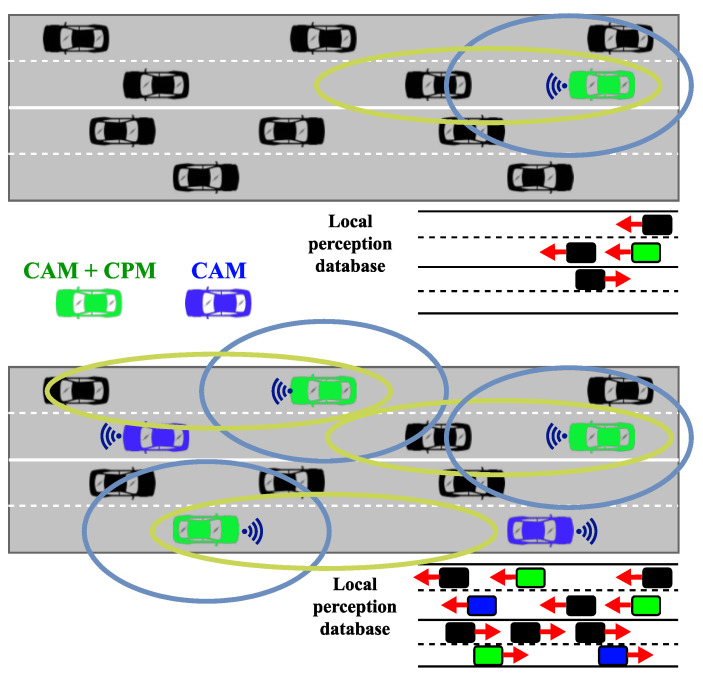
Change in the size of the perceivable environment using Collective Perception. The circles represent the detection ranges of different types of sensors equipped to the cars.

**Figure 2 sensors-23-07968-f002:**
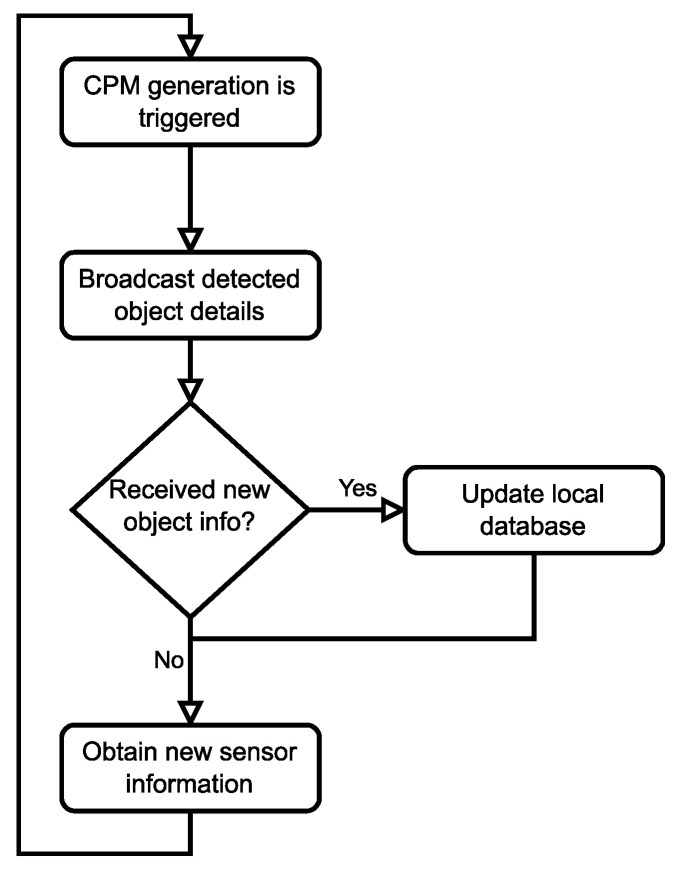
Periodic behavior of CPM generation.

**Figure 3 sensors-23-07968-f003:**
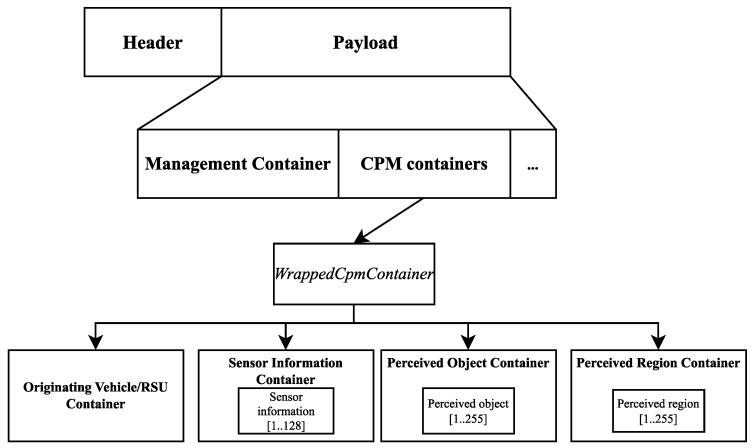
The general structure of a CPM.

**Figure 4 sensors-23-07968-f004:**
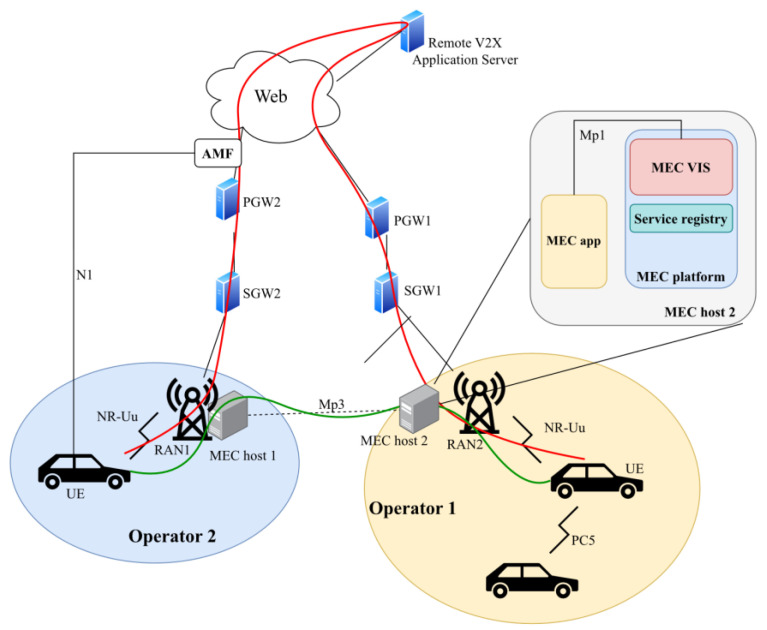
Data flow in a multi-MNO scenario in a traditional way (red) and using VIS (green).

**Figure 5 sensors-23-07968-f005:**
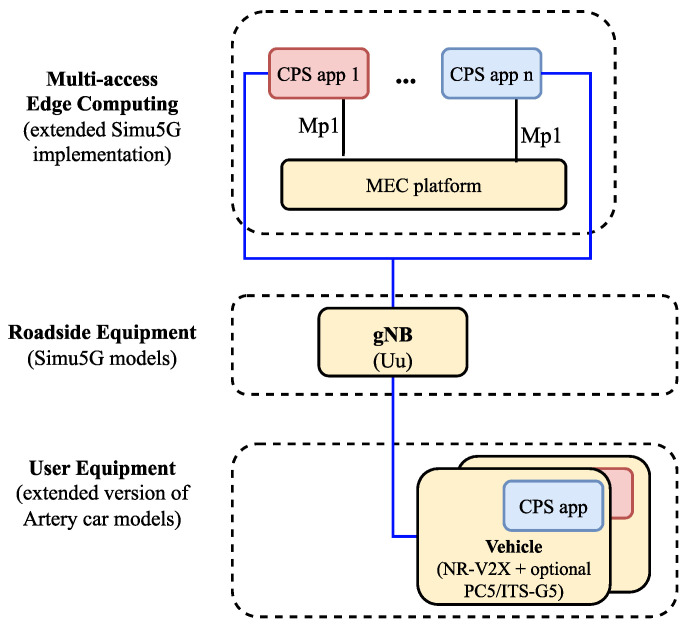
The system architecture of the simulation scenario.

**Figure 6 sensors-23-07968-f006:**
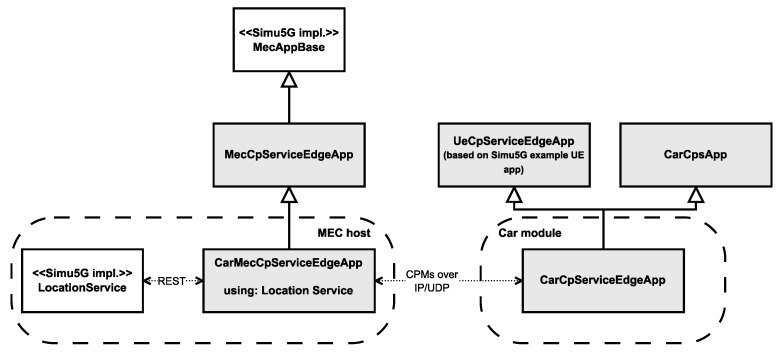
Simplified representation of the implemented application modules.

**Figure 7 sensors-23-07968-f007:**
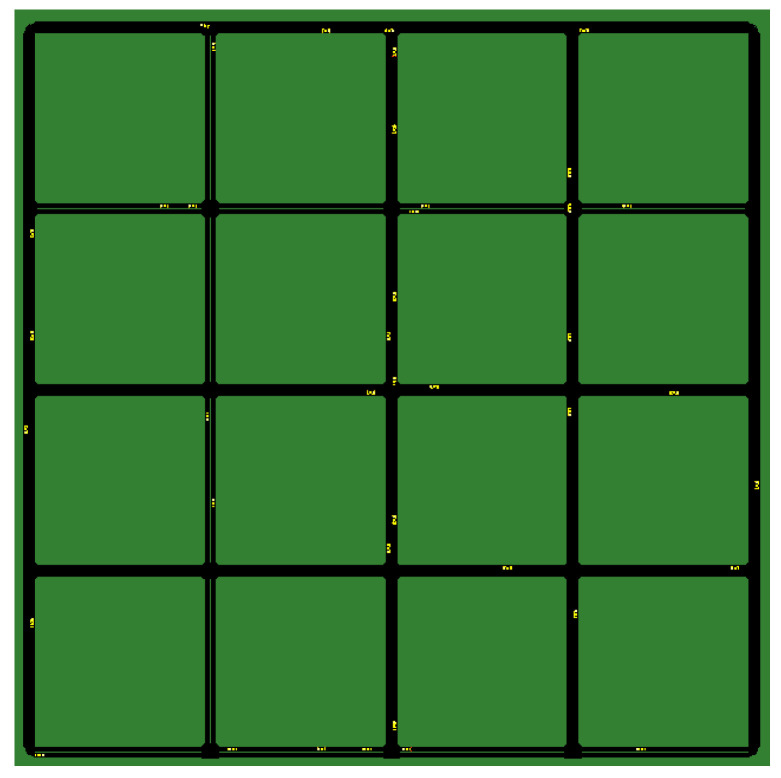
Four-by-four grid map used for the vehicle traffic simulation (with active cars).

**Figure 8 sensors-23-07968-f008:**
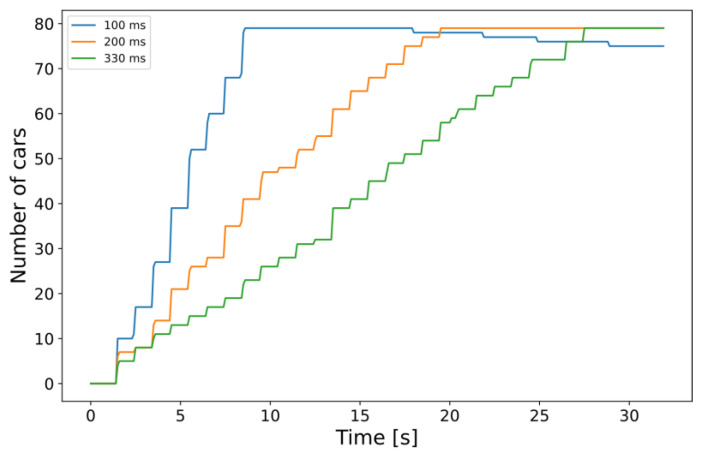
Number of cars simultaneously present in the simulation for different traffic densities. In each scenario, a new car is inserted, as the labels show (on average).

**Figure 9 sensors-23-07968-f009:**
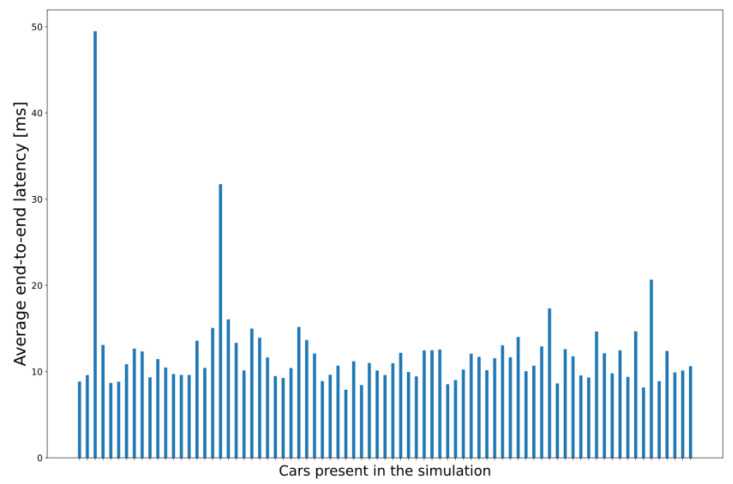
Average end-to-end CPM transmission latencies of each vehicle node in the simulation (required CPU metrics: 300 MIPS).

**Figure 10 sensors-23-07968-f010:**
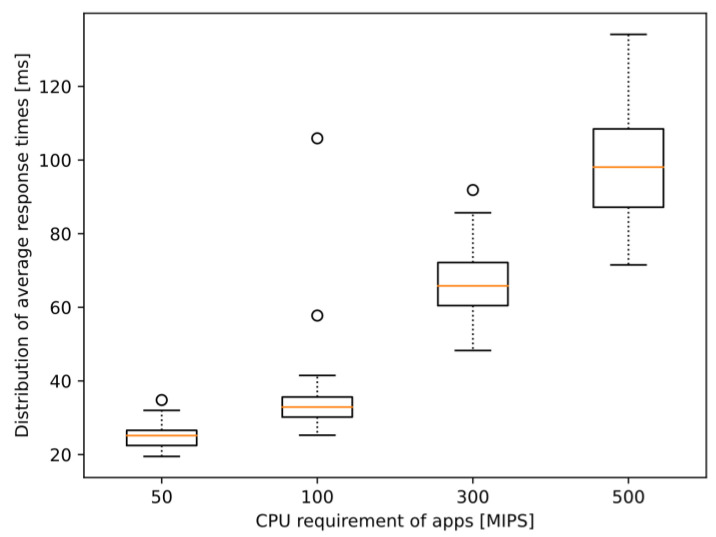
Distribution of average response times per car for different MEC application requirements in the scenarios of fixed CPU requirement model.

**Figure 11 sensors-23-07968-f011:**
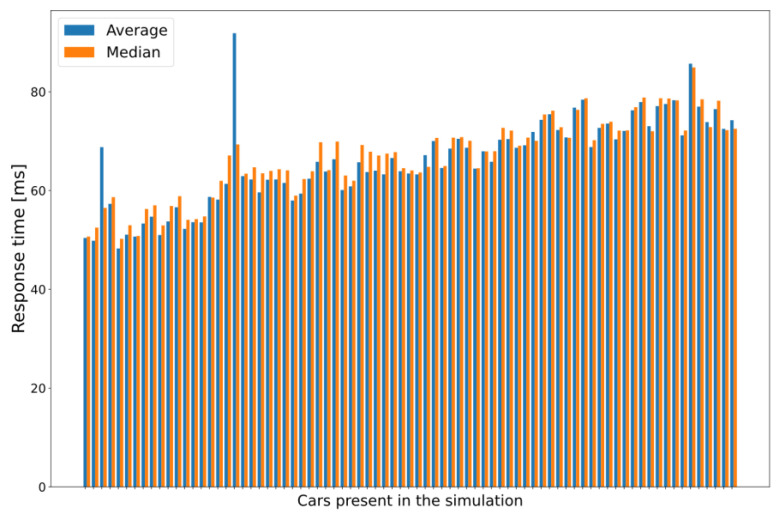
Average and median of the measured response times for each vehicle node (300 MIPS case).

**Figure 12 sensors-23-07968-f012:**
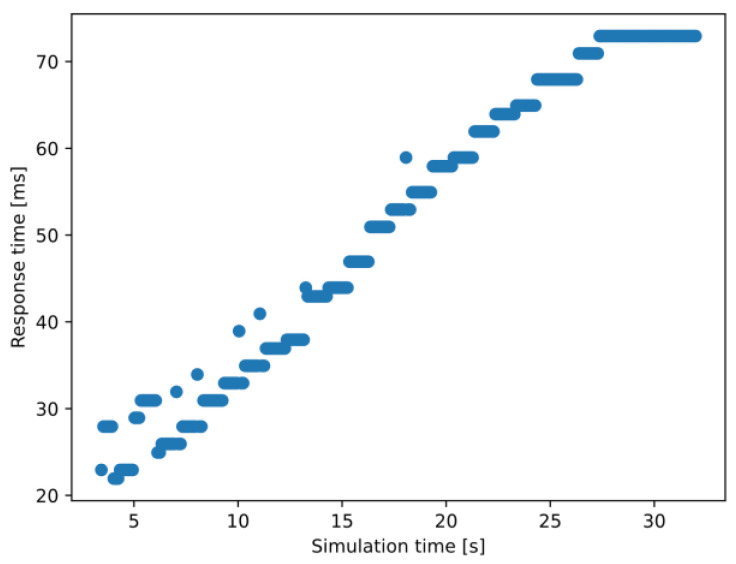
Change in the experienced response time values of a selected vehicle node during its presence in the simulation (300 MIPS case).

**Figure 13 sensors-23-07968-f013:**
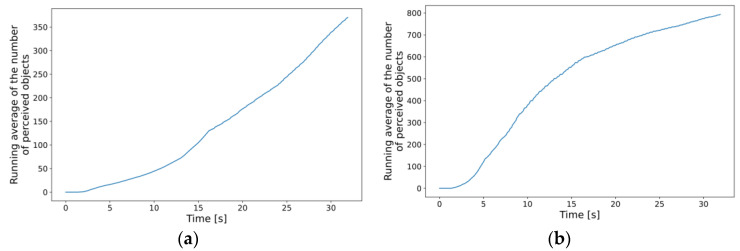
Running averages of the number of total perceived objects at the MEC aggregated level: (**a**) new car inserted every 330 ms on average; (**b**) new car inserted every 100 ms on average.

**Figure 14 sensors-23-07968-f014:**
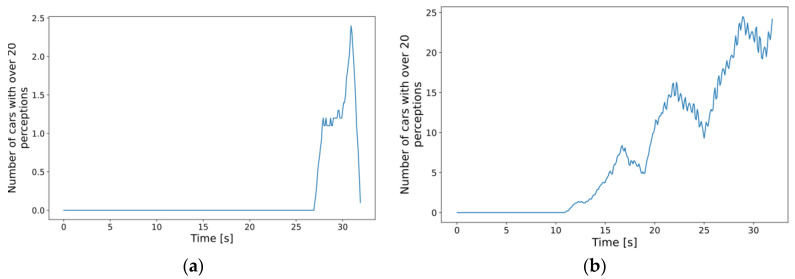
Moving window average (1 s) of the number of cars with more than 20 perceptions: (**a**) new car inserted every 330 ms on average; (**b**) new car inserted every 100 ms on average.

**Figure 15 sensors-23-07968-f015:**
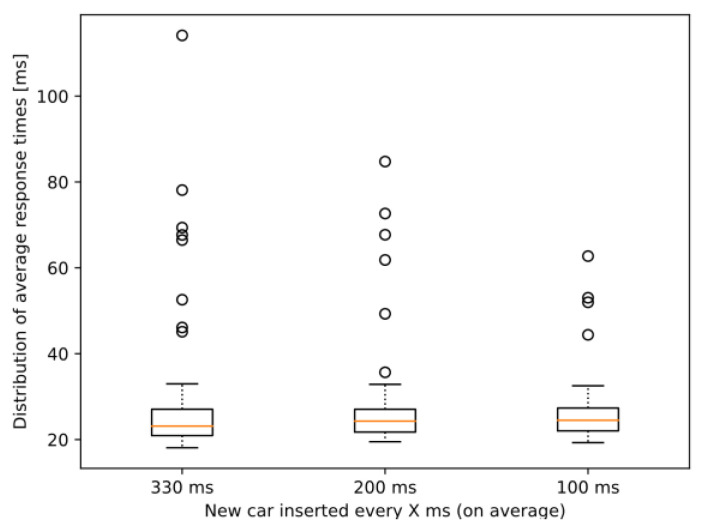
Distribution of average response times per car for different traffic densities in the scenarios of dynamic CPU requirement model.

**Table 1 sensors-23-07968-t001:** Comparison of V2X access technologies.

Parameters	802.11p	802.11bd	(LTE) C-V2X		5G NR-V2X	
			Short range (PC5 sidelink)	Long range (Uu)	Short range (PC5 sidelink)	Long range (Uu)
Modulation and coding scheme (MCS)	QPSK with BCC	BPSK up to 64-QAM with LDPC	QPSK to 64-QAM with turbo codes		up to 64-QAM with LDPC codes	
Doppler shift resistance methods	Preamble only	Preambles & Midambles	DMRS, 4/subframe		flexible DMRS	
Carrier frequency [GHz]	5.9	5.9	5.9	0.7, 0.8	5.9, 60 (mmWave)	available 5G bands
Sub-carrier spacing [kHz]	156.25	78.125, 156.25, 312.5	15		sub 6-GHz: 15, 30, 60	
					mmWave: 60, 120	N/A
PHY layer (waveform)	OFDM	OFDM	SC-FDMA		OFDM/DFTsOFDM	
Number of MCS	8	10	more than 20		more than 20	
Spatial streams	one	multiple	multiple		multiple	
Bandwidth [MHz]	10	10/20	Flexible: 1.4/5/20/20		sub 6-GHz: max. 100	
					mmWave: max. 400	N/A
Re-transmission	none	Congestion dependent	Blind		HARQ	
Communication types	broadcast	broadcast	broadcast	unicast	broadcast, groupcast, unicast	unicast
(Theoretical) transmission range [km]	about 1	about 1	2	up to 10	2	up to 10
Relative speeds [km/h]	252	500	500		500	

**Table 2 sensors-23-07968-t002:** Summary and comparison of selected scientific literature on MEC-performance analyses, testbeds, and simulation tools.

Research Effort	Year	Applied Framework	Contribution	Evaluation Target	Radio Technology
Sonmez et al. [43]	2017	CloudSim [44]	Model implementation	Edge computing simulator tool: EdgeCloudSim	n/a
Emara et al. [45]	2018	FlexRAN [46]/ OpenAirInterface	Model implementation and evaluation	End-to-end latency of MEC-assisted CAM service	LTE-Uu
Nardini et al. [47]	2020	Simu5G, Intel OpenNESS	Model proposal and evaluation	Capabilities of the MEC emulator	LTE/NR-Uu
Virdis et al. [48]	2020	Simu5G, Intel CoFluent	Model implementation and evaluation	End-to-end latency of different system deployments	LTE/NR-Uu
Massari et al. [49]	2021	NS-3 [50], 5G air simulator	Model implementation and evaluation	MEC simulator for Industry 4.0 scenarios	NR-Uu
Li et al. [51]	2021	MEC-Sim	Model implementation and evaluation	Evaluation of a proprietary MEC simulator solution	n/a
Passas et al. [52]	2021	OpenAirInterface [53]	Model implementation and evaluation	MEC service placement	LTE-Uu, Wi-Fi
Shi et al. [54]	2021	Intel CoFluent, SUMO	Model implementation and evaluation	End-to-end co-simulation framework	Simplified LTE
Rupp, Wischhof. [55]	2022	OMNeT++/ Simu5G	Service proposal and evaluation	Message prioritization strategy	NR-Uu
Schuhbäck et al. [56]	2023	CrowNet (OMNet+/Simu5G)	Model implementation and evaluation	Decentralized mobile crowd sensing strategy	LTE/NR Uu/PC5, DSRC, Wi-Fi
Noferi et al. [57]	2023	Simu5G	Model implementation and evaluation	MEC prototype, simulation framework	LTE/NR-Uu
Kovacs, Bokor [this article]	2023	Artery/Simu5G	Model implementation and evaluation	V2X stack simulator enhanced with 5G + MEC simulator implementation and CP service integration	LTE/NR Uu/PC5 (user plane), DSRC, Wi-Fi

**Table 3 sensors-23-07968-t003:** MEC resource parameters.

MEC Host Capability [MIPS]	MEC App CPU Requirement [MIPS]	Average End-to-End Latency [ms]	Average Response Time [ms]
400,000	50	11.4	25.0
400,000	100	12.5	34.1
400,000	300	12.0	66.0
400,000	500	11.8	97.5

## Data Availability

Not applicable.

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
