# Peer review of "Implementation of MEC-Assisted Collective Perception in an Integrated Artery/Simu5G Simulation Framework"

_sensors, 2023, doi:10.3390/s23187968_

Round 1

Reviewer 1 Report

Comments to the authors:

1. All the new points and the main difference between this paper and the previous works should be highlighted in this paper. Also, which new methods or technologies/solutions are proposed in this paper?

2. The main contribution of this paper should be showed more clearly, because the Reviewer feel that the contribution of the deployment and testing of V2X applications in this paper is quite limited.

3. There are many approaches which have been proposed before. Why the authors did not compare this work with these approaches to show the advantages (even disadvantages) of the proposed approach in this paper?

4. The authors should show more clearly that could the simulated results in this paper be applied in the practical applications? For example, do the simulated delay time, data rate, etc. satisfy the required QoS of the V2X systems in practice.

5. Almost references in this paper are new and updated. However, the number of references in this paper should be reduced.

6. Many contents in this paper introduce basic concepts, and therefore they should be shortened.

7. MCS in Table 1 is not defined. Please correct the similar typos.

8. Table 1 is over margin.

9. Equation in page 16 of 22 should be numbered.

10. There are still other typos in this paper which need to be corrected carefully.

Minor editing of English language is required

Reviewer 2 Report

An integrated framework for V2X simulations with 5G network elements is proposed in this paper. The topic is interesting and exhibits some novelty. However, the following major concerns should be addressed.

1. Related works are only listing the existing work without critically analyzing the pitfalls in existing work. In addition, motivations are not clear.

2. It is difficult to understand the actual contribution of the paper. The key contribution has not described in detail.

3. The authors should more clearly highlight the main contributions of this paper to ascertain the main technical contributions and improvements of this paper compared with previous work. Thus, the authors may provide a table in Section 3 summarizing the main differences/similarities of their paper with respect to the state-of-the-art.

4. Please add a figure in Section 2 to show the timeline of V2X communication.

5. Please fix grammatical mistakes and typo errors, e.g., “2.1.. Relevant V2X access technologies” - 2.1. Relevant V2X access technologies”

6. Some practical applications on V2X should be investigated and discussed.

a) "A V2I and V2V Collaboration Framework to Support Emergency Communications in ABS-Aided Internet of Vehicles," in IEEE Transactions on Green Communications and Networking, doi: 10.1109/TGCN.2023.3245098.

b) "Resource Allocation Modes in C-V2X: From LTE-V2X to 5G-V2X," in IEEE Internet of Things Journal, vol. 9, no. 11, pp. 8291-8314, 1 June1, 2022, doi: 10.1109/JIOT.2022.3159591.

Moderate editing of English language required.

Round 2

Reviewer 1 Report

The Reviewer agree with the authors' response. This paper can be accepted for the publication.

Minor editing of English language is required

Reviewer 2 Report

The reviewer's comments are fully addressed. This paper can be accepted.